# A Critical Review of Disinfection Processes to Control SARS-CoV-2 Transmission in the Food Industry

**DOI:** 10.3390/foods10020283

**Published:** 2021-01-31

**Authors:** Adrián Pedreira, Yeşim Taşkın, Míriam R. García

**Affiliations:** 1Bioprocess Engineering Group, IIM-CSIC, 36208 Vigo, Spain; apedreira@iim.csic.es; 2Lab of Recycling and Valorization of Waste Materials (REVAL), IIM-CSIC, 36208 Vigo, Spain; 3Food Engineering Department, Hacettepe University, Ankara 06800, Turkey; yesimtaskin@hacettepe.edu.tr

**Keywords:** SARS-CoV-2, COVID-19 pandemic, food industry, disinfection trade-offs, one-health

## Abstract

Industries of the food sector have made a great effort to control SARS-CoV-2 indirect transmission, through objects or surfaces, by updating cleaning and disinfection protocols previously focused on inactivating other pathogens, as well as food spoilage microorganisms. The information, although scarce at the beginning of the COVID-19 pandemic, has started to be sufficiently reliable to avoid over-conservative disinfection procedures. This work reviews the literature to propose a holistic view of the disinfection process where the decision variables, such as type and concentration of active substances, are optimised to guarantee the inactivation of SARS-CoV-2 and other usual pathogens and spoilage microorganisms while minimising possible side-effects on the environment and animal and human health.

## 1. Introduction

Efficient control of SARS-CoV-2 transmission is a high priority in the food industry. Food safety authorities agree that SARS-CoV-2, as other coronaviruses in previous outbreaks, is not transmitted through food consumption [1,2,3]. However, respiratory viral diseases are commonly spread via fomites (indirect route of transmission through inanimate objects or surfaces) [4,5] and different studies have proven that SARS-CoV-2 is stable for several days on different surfaces commonly used in the food industry [6,7,8,9,10]. The spread of the virus through this should be prevented by properly disinfecting and cleaning of the inanimate objects.

The food industry has been implementing cleaning and disinfection measures to effectively inactivate spoilage and pathogenic microorganisms like bacteria, yeasts, moulds and viruses [11], and hence avoid food waste or poisoning outbreaks. Cleaning must be done thoroughly before the disinfection step. In general, the efficacy of all disinfectants decreases in the presence of organic matter and the goal is to remove as much organic matter as possible by eliminating the gross dirt and by cleaning with water and surfactants [12]. Disinfection aims to achieve an acceptable standard of hygiene from the microbiological point of view [13].

Quite often the industry disinfects with aggressive substances at large concentrations and contact times to ensure good hygiene standards that are more than enough to inactivate SARS-Cov-2. For example, sodium hypochlorite has been historically employed to inactivate very resistant pathogens as vegetative bacteria and bacteria spores [14] and it has also shown efficacy against viruses, including SARS-CoV-2 [15].

Therefore, given the so-far known apparent simplicity to disinfect encapsulated viruses like SARS-CoV-2 [16,17] and the current evidence of limited transmission of the virus through fomites [18,19], should the food industry update the standard disinfection protocols to guarantee the product and workers safety?

A precautionary principle, given the 2020 pandemic, might point to applying over-conservative disinfection protocols with large amounts of active substances and contact times or, at least, to revise and adapt the protocols to ensure inactivation of SARS-CoV-2 [20]. Crowded and cooling conditions, common in slaughterhouses and the meat processing facilities, have been related with SARS-CoV-2 outbreaks. For example, from 9–27 April 2020, COVID-19 was diagnosed in 4913 workers from 113 meat and poultry processing facilities in the USA (approximately 3% of the workforce), with 20 COVID-19-related deaths [21]. In a more recent report, including the confirmed cases up to 31 May from a total of 239 facilities, the number of cases increases to 16,233 (3.1% to 24.5% infected workers per facility) and 86 COVID-19 related deaths [22]. Risk factors included prolonged closeness to other workers for long shifts, close contact during transportation in shared vans and contact through exposure to potentially contaminated shared surfaces such as break room tables or tools [8,23].

Nevertheless, over-use of disinfection to inactivate SARS-CoV-2 disregards any other effect on microorganisms present in the food processing plant, or the impact of the residual concentration of the chemicals on the environment and human and animal health. In this regard, it is considered critical the emergence and selection of antimicrobial-resistant bacteria due to the over-use of antimicrobial soaps and disinfectant cleaners. This concern has been discussed in the clinic sector [24], but not yet in the food industry.

In this review, we propose to analyse the disinfection processes using the “one health” approach, a holistic view considering the impact of the process beyond the objective of controlling the transmission of SARS-CoV-2. The review, therefore, is organised in three sections demanding to (1) understand which disinfectants or mixtures of disinfectants, concentrations and contact times are enough to inactivate SARS-CoV-2, (2) how these disinfectants work towards inactivating other food pathogens in the food industry and (3) their side-effects in the environment and the animal and human health due to misuse or overuse of disinfectants.

## 2. Disinfectants, Concentrations and Contact Times to Inactivate SARS-CoV-2 in the Food Industry

### 2.1. SARS-CoV-2 Stability on Surfaces

Despite the zoonotic origin of SARS-CoV-2, the virus is not a foodborne pathogen and to avoid its transmission in the food industry, it is critical to control transmission through fomites or contaminated objects [25]. The virus is stable on usual surfaces in the food industry, especially on non-porous surfaces, and remains infective for 3–7 days at room temperature in smooth surfaces such as glass, plastic and stainless steel [26]. By contrast, the virus has lower stability on cardboard (up to 2 days) and paper (up to 3 h) [15,27].

Previous reports about the prevalence of other coronaviruses on surfaces show similar behaviour. hCoV 229E and SARS-CoV-1, which share similar stability kinetics with SARS-CoV-2 [27], showed a prevalence in ceramic tiles of 4–5 days [28,29]. On dispensable materials such as latex gloves, cotton gowns and cotton gauze sponges, coronaviruses have lower persistence times (≤48 h) [30,31].

Food contact surfaces are typically made of materials where SARS-CoV-2 is stable for days, such as steel or plastic. Other common contact surfaces are made of materials like wood, rubber, ceramics or glass [32]. All surfaces must be smooth, non-porous and easily cleanable and free from large, randomly distributed irregularities such as pits, folds and crevices [33,34]. Regardless of the type of material, disinfection of objects and surfaces in contact with unprocessed foods is critical to avoid contaminated final products [35,36].

The stability time of SARS-CoV-2 on surfaces is affected by physico-chemical factors such as temperature, pH and humidity. Low temperatures extend virus viability from 7 (22 ${}^{\circ }$C) to 14 days (4 ${}^{\circ }$C), while stability is less than a day at 37 ${}^{\circ }$C and less than 5 min when exposed to 70 ${}^{\circ }$C [15]. SARS-CoV-2 shows a high persistence under pH ranging from 3–10 at room temperature. On the other hand, SARS-CoV-2 half-life at room temperature (24 ${}^{\circ }$C) decays from 18.6 h to 6.3 h when the relative humidity (RH) raises from 20% to 80% [37].

Table 1 summarises the stability of SARS-CoV-2, SARS-CoV-1 and other SARS-CoV surrogate coronaviruses on different surfaces at different temperatures and relative humidities. According to the literature, the stability of other coronaviruses on surfaces is analogous to SARS-CoV-2, being favoured by smooth surfaces and low temperature and RH. However, the lack of a standardized methodology makes complicated the comparison of literature results. From the author’s knowledge, to date just one work has compared the stability of SARS-CoV-2 with other coronavirus (SARS-CoV-1) under the same experimental conditions [27]. Another difficulty is the variability regarding the inoculum volume. Low volumes of virus inoculum are preferred since they probably provide a better simulation of the real mechanism of viral contamination of surfaces, such as sneezing, coughing or by touching with hands that have previously been in contact with mouth, eyes, nose or other contaminated surfaces.

### 2.2. Approved Substances for SARS-CoV-2 Disinfection

The food industry should ensure good hygienic practices and food safety management systems following the advice of international authorities, such as the Food and Agriculture Organizations of the United Nations, to prevent COVID-19 transmission [1,41].

For more specific information regarding the necessary standards for chemical disinfection of SARS-CoV-2, the industries comply with regulations of their respective competent authorities. Table 2 summarizes those active substances which can be employed in the formulation of disinfectants legally available against SARS-CoV-2 in the European Union and in the USA, as well as their status in the food industry of both territories.

The European Chemicals Agency (ECHA) is the competent authority in the European Union for determining legal disinfection substances. ECHA reports the lists of those substances approved or under review against SARS-CoV-2 for product-type 1 (human hygiene) and 2 (disinfectants and algaecides not intended for direct application to humans or animals), but not the list of substances for use in the feed and food area (product-type 4 or PT4) [42]. Therefore, the substances for disinfection of SARS-CoV-2 in the food industry should be in (1) the list of PT4 substances and in (2) any of the lists of substances approved for SARS-CoV-2.

The lists of authorised substances for SARS-CoV-2 are not a comprehensive representation of all the disinfectant substances legally employed in the EU, since compounds tagged as “under review” (e.g., ethanol) can be part of the formulation of authorized commercial products employed in some European countries if they are covered by their respective national legislation. ECHA also released a list of commercial products authorised under the Biocidal Products Regulation (BPR) with virucidal claims. This product list lacks information about the concentrations of active substances in the commercial products. Intending to provide a better indication of the market situation for disinfectant products, ECHA provides as an example information for some member states such as Spain [43] and The Netherlands [44].

The United States Environmental Protection Agency (EPA), the competent authority in the USA, publishes the so-called ”List N” gathering the authorized commercial disinfectants and active substances for use against SARS-CoV-2 [46] and their contact times. According to EPA, the authorised commercial products against SARS-CoV-2 must comply with at least one of the following criteria:(a)Demonstrate efficacy against SARS-CoV-2.(b)Demonstrate efficacy against a pathogen that is harder to inactivate than SARS-CoV-2.(c)Demonstrate efficacy against another type of human coronavirus similar to SARS-CoV-2.

The directions of use of disinfectants depend on the target pathogen and are specified on the product label guidance. For SARS-CoV-2, the “List N” informs if the directions to follow should be for SARS-CoV-2 or for another pathogen if it was not directly tested for SARS-CoV-2.

As in the case of ECHA, the list lacks information about the concentration of active substances in commercial products. Fortunately, this information has been gathered and published by the Emergency Care Research Institute (ECRI), a non-profit healthcare organization [45]. Figure 1 shows that most of the EPA authorised commercial products available in USA market were not tested directly for SARS-CoV-2 Figure 1. Instead, most authorised substances were tested following the b) criteria. However, the ranking of pathogens attending to the difficulty of being disinfected is based on debatable arguments, as we will discuss in Section 3.

Similarly to ECHA and EPA, there are other national authorities determining the status of disinfectants for SARS-CoV-2. In Australia, for example, the Therapeutic Goods Administration (TGA) has published a list with the commercial disinfectants in the Australian Register of Therapeutic Goods (ARTG) for use against SARS-CoV-2. Again, the authorised list does not provide information about the active ingredients nor the criteria followed to select the substances [47]. In Canada, the Health Products and Food Branch’s (HPFB) of the governmental department Health Canada (HC) has developed a list that is continuously updated with the likely effective commercial hard-surface disinfectants against SARS-CoV-2 and their active ingredients [48]. Substance concentrations are not reported but can be consulted in the Drug Product Database (DPD) [49]. The approval of a product is subject to the evidence provided by the manufacturers but inclusion criteria are not available for the general public.

### 2.3. Efficacy of Authorised Disinfectants

In a few months, researchers have been actively working to determine which disinfectants and concentrations are effective against SARS-CoV-2, but still more work is needed. The information is still scarce and substances are proved mostly in suspension test, instead of using dried viruses over surfaces better mimicking the conditions of the food industry [50,51].

Moreover, comparison between studies is usually difficult due to the lack of standardized testing procedures such as the high variability of contact times, and the differences in the kind and the concentration of the substances employed to simulate organic load. The initial viral concentration is also highly variable among studies, ranging from 10^5^ [52] to 10^8^ [15], and sometimes it is even omitted. This concentration is important since the minimum inhibitory concentration (MIC) may change with the number of organisms inoculated for different viruses, the so-called inoculum effect [53,54,55]. From the author’s knowledge, only one work has tested the virucidal activity of chemicals to SARS-CoV-2 using the international standards for disinfection on surfaces (ASTM E1052-20) and suspension (EN 14476:2013) [56].

#### 2.3.1. Disinfectants Tested against SARS-CoV-2 or Similar Coronaviruses

The list of tested disinfectants against SARS-CoV-2 and SARS-Cov-2 surrogates changes continuously. We have gathered the information published until September 2020 in a table that can be consulted in the following link https://doi.org/10.5281/zenodo.4297322. This data-sheet covers those disinfectants tested against SARS-CoV-2 or other coronaviruses. Data were extracted from several research articles indicated in the reference row. The data-sheet comprises a total of 11 fields with info regarding the virus (virus and strain/isolate names), formulation (substance(s) and its concentration in percentage) and test characteristics (suspension or surface tested, kind of surface, use dilution, disinfectant and inoculum volumes, organic load type and concentrations and contact time) as well as their results, normalized in terms of log_10_ viral infectivity reduction.

Just a few common disinfectants have been proven effective against SARS-CoV-2 at short contact times: 30–80% ethanol, 30–75% propanol, 0.45–7.5% povidone-iodine and 5–6% sodium hypochlorite (household bleach) (See table in https://doi.org/10.5281/zenodo.4297322). All of them are allowed in the food industry and have been successfully tested against SARS-CoV-2, reaching a fast reduction on viral infectivity of more than 4 orders of magnitude (99.99% reduction), the level suggested as effective by the German Association for the Control of Virus Diseases and the Robert Koch Institute [57].

Other common disinfectants such as 0.1% benzalkonium chloride (BAC) were effective against SARS-CoV-2, but at the expense of large contact times [56]. In fact, previous tests with quaternary ammonium compounds (QACs), such as BAC, and other coronaviruses also showed the need for large contact times [58,59]. More specifically, two commercial disinfectant products were tested: (1) Mikrobac Forte composed of BAC and dodecylbispropylene triamine and (2) Kohrsolin FF with BAC, didecyldimethylammonium chloride (DDAC) and glutaraldehyde as active ingredients. Results showed that, after 30 min, the disinfection was not as effective as using 80% ethanol with only 30 s of exposure [38]. Tests using BAC to disinfect SARS-CoV-1 surrogate coronaviruses (such as human coronavirus causing common cold [59,60], canine coronavirus and mouse hepatitis virus [58]) showed similar results to ethanol and propanol, although again with exposition times of minutes (5–10 min).

More controversial is the efficacy of other common disinfectants in the food industry such as hydrogen peroxide. Although enveloped viruses are more sensitive to hydrogen peroxide than non-enveloped ones [61], all seem to point to a low performance of hydrogen peroxide against coronaviruses. Hydrogen peroxide is minimally effective against SARS-CoV-2, reaching just a poor viral infectivity reduction of 1–1.8 log_10_ in a work concentration of 1–6% after 30 s of exposition [52]. Previous reports showing a good performance of hydrogen peroxide against coronaviruses are subordinate to long exposure times or the presence of other active ingredients in the formulation. In hCoV 229e, a satisfactory ≥4.00 log_10_ viral infectivity reduction was achieved after 60 s of exposition to 0.5% hydrogen peroxide [62]. However, an unknown percentage of non reported food-grade surfactants were also present in the formulation. In TGEV, a viral infectivity reduction ∼5 log_10_ was achieved employing vaporized 35% hydrogen peroxide [63]. Unfortunately, the long contact time (2–3 h) makes its application very cost–time expensive. The World Health Organization (WHO) Formulations I and II, which contain 0.125% hydrogen peroxide in their composition besides 80% ethanol (Formulation I) and 70% 2-propanol (Formulation II), have very good results against coronaviruses but do not allow to ensure the hydrogen peroxide performance when it acts alone. In fact, both 30% ethanol and 30% propanol, used independently reached the same inactivation levels [64].

It should be stressed that there are also substances legally commercialized against SARS-CoV-2 without proven efficacy. For example, 2% polyhexamethylene biguanide (PHMB) [65] and salicylic acid 0.10% [56] have failed in SARS-CoV-2 inactivation, not reaching the recommended log_10_ viral reduction ≥4.00. However, PHMB is found in commercial products against SARS-CoV-2 authorized by the EPA and it has been classified as an authorized substance with the same purpose in the ECHA list, which also includes salicylic acid.

Interestingly, disinfectants such as 4.7% chloroxylenol [56] and 0.05% clorhexidine [15] show good performance against SARS-CoV-2, but have not been approved by the EPA nor ECHA to SARS-CoV-2 disinfection. A reason for this contradiction could be that, for example, chlorhexidine is not efficient in the inactivation of other coronaviruses [58].

So far, no other substances have been tested against SARS-CoV-2 and their use is just supported by previous reports showing antiviral activity against other viruses from the Coronaviridae family. This is the case of DDAC, glutaraldehyde and phenolic compounds. DDAC has never been tested against coronaviruses as a single active ingredient, but in mixtures with other disinfectants [66]. Glutaraldehyde 0.5% gets a ≥4.01 log_10_ reduction in SARS-CoV-1 after 120 s of exposition [38]. Other studies employing glutaraldehyde 2% reported viral reductions >3.00 log_10_ after 60 s in canine coronavirus [67] and 5 log_10_ after 24 h in hCoV [60]. Several phenolic formulations reached a >3.00 log_10_ viral reduction against hCoV [60] and a mixture comprising o-phenylphenol 9.09% and p-tertiary amylphenol 7.66% obtained a reduction of 2.03 log_10_ in transmissible gastroenteritis virus and 1.33 log_10_ in mouse hepatitis virus [68].

#### 2.3.2. Disinfectants Tested against Viruses Different from Coronavirus

There is a concerning large number of authorized commercial products for SARS-CoV-2 disinfection which contain active ingredients that have not been tested against the COVID-19 causing virus nor other coronaviruses. This case includes organic acids as well as thymol.

Organic acids have not been tested against SARS-CoV-2, with the exception of salicylic acid explained in the previous section [56], or other coronaviruses, and their virucidal effect against other viruses is controversial. Formic acid showed to be ineffective against the non enveloped viruses mammalian orthoreovirus type 1 and bovine adenovirus type 1 [69]. In another study with enveloped viruses, formic and citric acids were not effective against the bovine viral diarrhoea virus, although they showed good results in the case of vaccinia virus, both enveloped viruses [70]. It should be noted that in these studies the exposition times were long (10–30 min), avoiding to know its cost–time efficiency. Moreover, taking into account that the action mechanism of organics acids is related to the decrease in the extra-viral pH which alters the steric disposition of specific receptors on the surface [71], the high stability of SARS-CoV-2 in acidic conditions points to a subsequent resistance to organic acids. Despite all this, organic acids as citric, lactic, formic and performic acid are active ingredients of SARS-CoV-2 disinfectants authorized by the EPA and the ECHA.

The use of thymol is authorized by the EPA but not by the ECHA. The reports about its virucidal effectiveness are scarce and restricted to herpes virus [72,73], without any reference against other viruses. Among the authorized commercial products against SARS-CoV-2, substances mixtures are very common. Although a synergy or additive effect could be expected in a components mixture, this fact should be previously evaluated [20]. As an example, WHO Formulations I and II, which contain 0.125% hydrogen peroxide besides 80% ethanol (Formulation I) and 70% 2-propanol (Formulation II), have very good results against coronaviruses (see table in https://doi.org/10.5281/zenodo.4297322), but 30% ethanol and 30% propanol used independently reached the same inactivation levels [64].

## 3. Potential of SARS-CoV-2 Disinfectants to Inactivate Other Pathogens or Spoilers in the Food Industry

The food industry should guarantee that other pathogens, or even food spoilage microorganisms, in addition to SARS-CoV-2 are inactivated. Adequate precautions should be taken to prevent food from being contaminated by disinfectants, during cleaning or disinfection of rooms, equipment or utensils [74]. Thus, the concentration of the active substance, as well as dilutions (if needed) and exposure times, have to be calculated taking into account the presence of other microbes.

According to the modern interpretation of the Spaulding classification [16,17], enveloped viruses such as coronaviruses are weak and easy to eradicate in comparison to other pathogens because the disruption of their lipidic envelope is considered enough to render them non-infectious. The Spaulding classification provides a very useful overview but disregards many factors affecting disinfection depending on the type of microorganisms. Some factors are the organic load concentration and type of surfaces and the clumping and biofilm formation of virus and bacteria, respectively [17]. In addition, useful parameters to consider when disinfecting, such as the persistence time on surfaces, do not follow the ranking in Spaulding classification. For example, SARS-CoV-2 persistence on certain surfaces is higher or similar than the persistence of harder-to-eradicate Gram-positive bacteria (*Vibrio cholerae* 1–7 days, *Helicobacter pylori* ≤ 90 min) and non-enveloped viruses (norovirus 8 h–7 days, echovirus 7 days and papovavirus 8 days) [11].

Even if those affecting factors are not in place, for the optimal disinfection of different usual microorganisms in the food industry together with SARS-CoV-2, the Spaulding classification is only a first approximation and more information is usually needed. On the one hand, and in agreement with the classification, disinfectants might only work on more susceptible groups. For example, 0.1% BAC inactivates SARS-CoV-2 but is not sufficiently effective for usual resistant bacterial strains found in the food industry such as *Staphyloccocus* spp., *Klebsiella* spp. and *Escherichia coli* [15,56]. However, the use of strong disinfectants to kill hard-to-eradicate groups does not always assure the disinfection of susceptible groups as Spaulding classification has many exceptions. For example, hydrogen peroxide, considered a high-level common disinfectant [17], is not very effective against SARS-CoV-2 [52].

Attending to the information gathered, those disinfectants that have been clearly proven effective against SARS-CoV-2, i.e., ethanol, propanol, povidone-iodine and sodium hypochlorite, have also been historically employed for other pathogens as vegetative bacteria, bacteria spores, fungi, enveloped and non enveloped viruses (including different coronaviruses) with remarkably efficacy [14]. Therefore a standard two-stage disinfection comprising a first cleaning stage with detergent followed by the application of a disinfectant should be enough to inactivate SARS-CoV-2 if proper disinfectant, concentration and exposure times are employed.

On the other hand, alternative substances that have not been tested on coronaviruses or show a poor performance against SARS-CoV-2, such as organic acids, thymol and PHMB, have also a controversial efficacy against other pathogens and are not recommended. The activity of organic acids is highly dependent of the pH and they are considered as bacteriostatic/fungistatic substances instead of bactericide/fungicide [75]. In addition, acidophilic organism as acetic bacteria has a high tolerance to organic acids and some bacteria, fungi and yeasts can even employ them as carbon source. Likewise, thymol seems not to be active against *Pseudomonas* spp., a food-spoilage and pathogen bacteria genus commonly present in food industry [76,77,78] and PHMB is able to inactivate vegetative bacteria and yeast but it has not been demonstrated for sporicide nor sporostatic activity [79].

## 4. Disinfection under the Paradigm of the “One-Health” Approach

Although necessary, disinfection has side-effects that should be minimized by proper selection of disinfectants and concentrations. Understanding disinfection trade-offs is particularly relevant within the urgency of a pandemic where over-conservative and aggressive disinfection may be preferred due to the lack of solid evidence about the new pathogen in the literature.

The majority of chemicals used in disinfection are harmful or corrosive, being dangerous in application [80,81]. As a matter of fact, in the first month of 2020 the number of daily exposures to cleaners and disinfectants reported to U.S. poison centres increased substantially [82]. Workers in the food industries should be properly trained in this regard. However, under the high demand for disinfectants, there might be a shortage of required active compounds [83], leading to the employment of more aggressive substances.

Moreover, non-volatile disinfectants might reach the environment and cause harm at different levels. Inactivation of bacteria involved in relevant transformations may disrupt ecosystems, such as the case of bacteria transforming nitrogenous compounds [84]. Non-volatile compounds may remain on solid surfaces and, after water rinsing or natural precipitation, may contaminate food [85] or move to soils or water [81]. QACs, for example, are toxic to aquatic organisms [86] and have been related with a decrease of mouse fertility [87]. Chlorinated disinfectant residues may be also lethal for aquatic organisms, but also toxic to terrestrial animals (birds and mammals) [81]. Silver exposition may result in toxic effects mostly for skin and liver, given sufficient dosage and lengths of exposition [88,89].

Disinfection by-products (DBPs) may be also a concern. Due to water sanitation, chlorination by-products, such as chloroform, trihalomethanes and haloacetic acids, have been extensively studied with many proven associated risks [84].

Last but not least overuse or misuse of certain disinfectants in the food industry may promote resistance acquisition or selection, which constitutes a major threat to human and animal health [90,91]. Phenolics [92,93], glutaraldehyde [94,95] and silver [89,96], for example, are allowed substances to inactivate SARS-Cov-2 with proven risks of promoting resistance and cross-resistance.

QACs, however, are probably the most alarming substances in the food industry because their widespread employment and potential to generate resistance or even cross-resistance with antibiotics [91,97,98]. Resistance to benzalkonium chloride, the most common QAC, has been measured in 57 bacteria species [99]. Certain strains even have MIC values several times higher (1000–3000 mg/L) than the BAC concentration in commercial disinfectants covered by ECHA and EPA, making them ineffective in their recommended dosage. Although other mechanism can be involved, it is assumed that QAC resistance is mainly mediated by *qac* genes, responsible of the synthesis of efflux pumps. These multidrug efflux pumps are non-specific detoxification mechanism and, thus, sub-MIC QAC exposition can also lead to the promotion of cross-resistance to dissimilar biocides and antibiotics. In relation to the food industry, *qac* genes have been found in several *E. coli* strains isolated from retail meats in the USA [100] and China [101]. In addition, BAC exposition increased its tolerance in 76 bacterial strains isolated from food, reducing their susceptibility to other biocides (hexachlorophene, DDAC, triclosan and chlorhexidine) and antibiotics such as ampicillin, sulfamethoxazol and cefotaxime [102], and conferred different degrees of resistance against oxytetracycline, amoxicillin, ampicillin, levofloxacin and gentamicin in *Salmonella* sp. isolated from supermarket meat [103].

Fortunately, no evidences of genetic resistance have been reported to ethanol, propanol, povidone-iodine and sodium hypochlorite, the most effective disinfectants against SARS-CoV-2. Moreover, broad sense resistance to these substances is, when existing, derived from biofilm formation and thus easily avoidable if proper cleaning is carried out before disinfection [75,104]. However, they have also disadvantages. A repeated use of alcohols over certain inanimate surfaces can damage the material. Iodine can cause brown colour stains on certain porous materials such plastics and clothing. Free chlorine is an aggressive chemical that can promote corrosion of metal surfaces, especially at higher concentrations. Chlorine gas can be released from chlorine solutions such as sodium hypochlorite solutions when exposed to heat or acid substances commonly found in domestic and industrial cleaners [14].

## 5. Conclusions

At the beginning of the pandemic, some works proposed to update the disinfection protocols in the food industry to control the new pathogen by considering aggressive chemical treatments. As more evidence became available, this necessity was not clear due to (1) the limited evidence of transmission through fomites, (2) the proven efficacy of the standard disinfection protocols in the food industry to inactivate SARS-CoV-2 and (3) the possible impact of aggressive disinfection on the environment and human and animal health, in alignment with the one-health perspective, especially because of the potential emergence of bacterial resistance.

The following substances, based on current literature and regulations, are recommended for disinfection of SARS-CoV-2: ethanol, propanol, povidone-iodine and sodium hypochlorite. They are the most effective disinfectants against SARS-CoV-2 showing, at the same time, minor side-effects. Other disinfectants regulated against SARS-CoV-2, such as some QACs, required longer contact times and may induce bacterial resistance and cross-resistance or being fluxed to the environment causing harms to different ecosystems. In general, non-volatile disinfectants should be avoided. These substances might reach the environment and disrupt ecosystems or even remain on solid surfaces and contaminate food after water rinsing.

More research is needed to optimally select the type of disinfectant, its concentration and its contact time to inactivate major pathogens, including SARS-CoV-2, while minimizing the impact on the environment and animal and human health. The effective concentrations of disinfectants are difficult to assess due to the lack of standardization among different studies. On the other hand, only the impact of the disinfectants used on water treatment are well studied, but not the active compounds used to disinfect objects or surfaces in the food industry.

## Figures and Tables

**Figure 1 foods-10-00283-f001:**
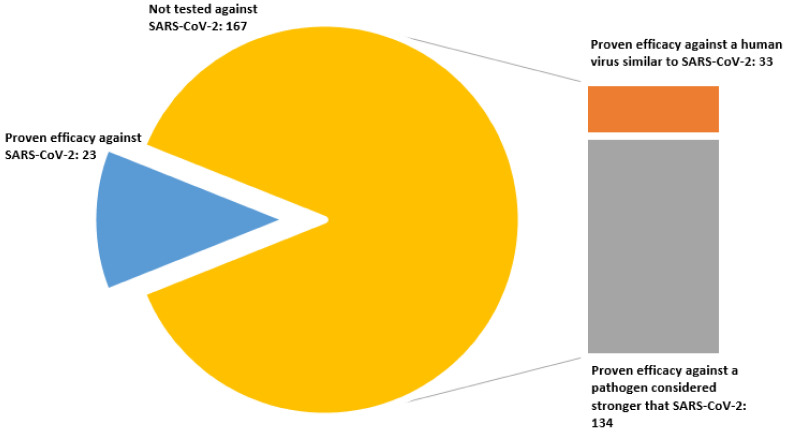
Commercial products authorized to be employed in the food industry (either requiring post-rise or not) against SARS-CoV-2 according to the EPA List N grouped by their acceptance criteria.

**Table 1 foods-10-00283-t001:** Persitence time of SARS-CoV-1, SARS-CoV-2, MERS-CoV and hCoV over different surfaces. TCDI50: Median tissue culture infectious dose, RH: relative humidity, ND: not defined, PVC: polivynil chloride, PE: polyethylene, PTFE: polytetrafluoroethylene, PP: polypropropylene, RC: room conditions. ^(a)^: Calculated from plaque forming units (PFU) following the instructions (1 TCID_50_ ≈ 0.7 PFU) reported by the American Type Culture Collection (ATCC). ^(c)^: Personal communication from the authors.

Virus	Surface/Fomite	Inoculum	TCDI_50_	T (${}^{\circ }$C)	RH (%)	Stability	Ref.
hCoV 229E	Aluminium	10 L	5.5 × 10^5^	21	55–70	>8 h	[31]
	Brasses >70% Cu	20 L	1.4 × 10^3^ ^(a)^	21	30–40	≤40 min	[29]
	Brasses ≥70% Cu	1 L	1.4 × 10^3^ ^(a)^	21	30–40	≤5 min	[29]
	Ceramic tile	20 L	1.4 × 10^3^ ^(a)^	21	30–40	>120 h	[29]
	Copper	1 L	1.4 × 10^3^ ^(a)^	21	30–40	≤5 min	[29]
	Copper-nickel ≥79% Cu	20 L	1.4 × 10^3^ ^(a)^	21	30–40	≤50 min	[29]
	Copper-nickel 70% Cu	20 L	1.4 × 10^3^ ^(a)^	21	30–40	≤2 h	[29]
	Cotton gauze sponges	10 L	5.5 × 10^5^ ^(a)^	21	55–70	≤6 h	[31]
	Glass	20 L	1.4 × 10^3^ ^(a)^	21	30–40	>120 h	[29]
	Latex gloves	10 L	5.5 × 10^5^	21	55–70	≤6 h	[31]
	Plastic (PVC)	20 L	1.4 × 10^3^ ^(a)^	21	30–40	>120 h	[29]
	Plastic (PE)	500 L	10^7^	21–25	ND	>72 h	[38]
	Plastic (PTFE)	20 L	1.4 × 10^3^ ^(a)^	21	30–40	>120 h	[29]
	Silicon rubber	20 L	1.4 × 10^3^ ^(a)^	21	30–40	≤120 h	[29]
	Steel	20 L	1.4 × 10^3^ ^(a)^	21	30–40	>120 h	[29]
hCoV OC43	Aluminium	10 L	5.5 × 10^5^	21	55–70	≤3 h	[31]
	Cotton gauze sponges	10 L	5.5 × 10^5^	21	55–70	≤1 h	[31]
	Latex gloves	10 L	5.5 × 10^5^	21	55–70	≤1 h	[31]
MERS-CoV EMC/2012	Plastic (undefined)	100 L	10^6^	20	40	≤72 h	[39]
	Plastic (undefined)	100 L	10^6^	30	30	≤48 h	[39]
	Plastic (undefined)	100 L	10^6^	30	80	≤24 h	[39]
	Steel	100 L	10^6^	20	40	≤72 h	[39]
	Steel	100 L	10^6^	30	30	≤48 h	[39]
	Steel	100 L	10^6^	30	80	≤24 h	[39]
SARS-CoV-1 HKU39849	Plastic (Undefined)	10 L	10^7^	22–25	80	>120 h	[40]
SARS-CoV-1 P9	Ceramic tile	300 L	10^6^	20	ND	≤96 h	[28]
	Glass	300 L	10^6^	20	ND	≤120 h	[28]
	Metal (undefined)	300 L	10^6^	20	ND	>120 h	[28]
	Paper (filter paper)	300 L	10^6^	20	ND	>120 h	[28]
	Paper (press paper)	300 L	10^6^	20	ND	≤120 h	[28]
	Plastic (undefined)	300 L	10^6^	20	ND	≤120 h	[28]
	Wood	300 L	10^6^	20	ND	≤96 h	[28]
SARS-CoV-1 GVU6109	Cotton gown	5 L	10^4^	37	ND	≤5 min	[30]
	Cotton gown	5 L	10^5^	37	ND	≤1 h	[30]
	Cotton gown	5 L	10^6^	37	ND	≤24 h	[30]
	Paper (request form)	5 L	10^4^	37	ND	≤5 min	[30]
	Paper (request form)	5 L	10^5^	37	ND	≤3 h	[30]
	Paper (request form)	5 L	10^6^	37	ND	≤24 h	[30]
	Plastic disposable gown	5 L	10^4^	37	ND	≤1 h	[30]
	Plastic disposable gown	5 L	10^5^	37	ND	≤24 h	[30]
	Plastic disposable gown	5 L	10^6^	37	ND	≤48 h	[30]
SARS-CoV-1 FFM1	Plastic (PE)	500 L	10^7^	21–25	ND	≤216 h	[38]
SARS-CoV-1 Tor2	Capboard	50 L	10^7^	RC ^(c)^	RC ^(c)^	≤8 h	[27]
	Copper	50 L	10^7^	RC ^(c)^	RC ^(c)^	≤8 h	[27]
	Plastic (PP)	50 L	10^7^	RC ^(c)^	RC ^(c)^	≤72 h	[27]
	Steel	50 L	10^7^	RC ^(c)^	RC ^(c)^	≤72 h	[27]
SARS-CoV-2 WA1-2020	Capboard	50 L	10^7^	RC ^(c)^	RC ^(c)^	≤24 h	[27]
	Copper	50 L	10^7^	RC ^(c)^	RC ^(c)^	≤4 h	[27]
	Plastic (PP)	50 L	10^7^	RC ^(c)^	RC ^(c)^	≤72 h	[27]
	Steel	50 L	10^7^	RC ^(c)^	RC ^(c)^	≤72 h	[27]
SARS-CoV-2 Unknow strain	Paper (tissue paper)	5 L	10^7^–10^8^	22	65	≤3 h	[15]
	Paper (undefined)	5 L	10^7^–10^8^	22	65	≤3 h	[15]
	Wood	5 L	10^7^–10^8^	22	65	≤3 h	[15]
	Cloth	5 L	10^7^–10^8^	22	65	≤48 h	[15]
	Glass	5 L	10^7^–10^8^	22	65	≤96 h	[15]
	Paper (banknote)	5 L	10^7^–10^8^	22	65	≤96 h	[15]
	Steel	5 L	10^7^–10^8^	22	65	≤168 h	[15]
	Plastic (undefined)	5 L	10^7^–10^8^	22	65	≤168 h	[15]
	Mask (inner layer)	5 L	10^7^–10^8^	22	65	≤168 h	[15]
	Mask (outter layer)	5 L	10^7^–10^8^	22	65	>168 h	[15]

**Table 2 foods-10-00283-t002:** Status of the substances against SARS-COV-2 according to The European Chemicals Agency (ECHA) and Environmental Protection Agency (EPA). CP: commercial products; AP: approved; UR: under review; NE: not evaluated. a: Substance not found in the formulation of commercial biocides authorized by ECHA to be employed in the food and feed areas (PT 4). b: Substance not found in the formulation of EPA authorised commercial biocides against SARS-CoV-2 (not area distinction). Compounds tagged as “under review” in ECHA column can be part of the formulation of authorized commercial products employed in some European countries if they are covered by their respective legislations [42]. Substances concentrations in commercial products available in the EU were obtained from the state members of Spanish [43] and Dutch [44] markets, both selected as reference markets by ECHA [42]. Substance concentrations in commercial products available in the USA were extracted from the Emergency Care Research Institute (ECRI) [45]. * N-C10-16-alkyltrimethylenedi- reaction products with chloroacetic acid. ** Mixture of 5-chloro-2-methyl-2H-isothiazol-3-one (EINECS 247-500-7) and 2-methyl-2H-isothiazol-3-one (EINECS 220-239-6).

	SARS-2-CoV Status		Food Industry Status		Concentration (%) in CP	
	ECHA	EPA	ECHA	EPA	EU	USA
**Alcohols**						
Ethanol	AP	AP	UR	AP	65.00–75.00	7.50–68.61
1-propanol	AP	NE	AP	AP	17.00–49.00	b
2-propanol	AP	AP	AP	AP	9.99	12.25–63.25
**Aldehides**						
Glutaraldehyde	AP	AP	AP	AP	2.50–12.00	7.00
Glyoxal	AP	NE	AP	AP	6.00	b
**Amines**						
Ampholyt 20 *	AP	NE	AP	NE	a	b
Glucoprotamin	AP	NE	AP	AP	a	b
N-(3-aminopropyl)- N-dodecylpropane-1,3-diamine	AP	AP	AP	AP	0.1–2.80	7.50–68.61
**Biguanides**						
Polyhexamethylene biguanide hydrochloride	AP	AP	AP	AP	a	0.050–0.089
**Chlorine based compounds**						
Calcium hypochlorite	AP	NE	AP	AP	a	b
Chlorine dioxide	UR	AP	UR	AP	a	0.20–5.00
Hypochlorous acid	UR	AP	UR	AP	0.017	0.017–0.046
Sodium hypochlorite	AP	AP	AP	AP	2.60–13.00	0.086–8.60
Sodium chlorite	UR	AP	UR	AP	a	0.50–30.50
Sodium dichloroisocyanurate	UR	AP	UR	AP	81.00	7.00–48.21
Tetrachlorodecaoxide complex	UR	NE	UR	NE	a	b
Tosylchloramide sodium	UR	NE	UR	AP	a	b
Trichloroisocyanuric acid	UR	NE	UR	AP	a	b
**Iodine and iodophors**						
Iodine	AP	NE	AP	AP	a	b
Povidone-iodine	AP	NE	AP	AP	a	b
**Isothiazolinones**						
Mixture of CMIT/MIT **	AP	NE	AP	AP	a	b
**Organic acids**						
Citric acid	AP	AP	AP	AP	a	0.60–6.00
Formic acid	UR	NE	UR	AP	a	b
Glycolic acid	UR	AP	UR	AP	a	11.19
Lactic acid	AP	AP	AP	AP	0.42–1.75	0.16–34.10
Performic acid	UR	NE	UR	AP	a	b
**Peroxides and derivates**						
Hydrogen peroxide	AP	AP	AP	AP	0.20–35.00	0.30–27.50
Peracetic acid	AP	AP	AP	AP	0.05–5.00	0.05–15.00
Peroxyoctanoic acid	UR	AP	UR	AP	a	0.63
Potassium peroxymonosulfate	NE	AP	UR	AP	49.70	21.41
Sodium carbonate peroxyhydrate	NE	AP	NE	AP	a	12.10–29.75
**Phenolic compounds**						
2-Phenylphenol	UR	AP	UR	AP	a	0.026–10.50
2-Benzyl-4-chlorophenol	NE	AP	NE	AP	a	0.023–3.03
4-tert-amylphenol	NE	AP	NE	AP	a	5.27–7.66
5-chloro-2-(4-chlorophenoxy) phenol	AP	NE	AP	NE	a	b
Biphenyl-2-ol	AP	AP	AP	AP	a	0.06
Salicylic acid	UR	NE	UR	AP	a	b
Thymol	NE	AP	NE	AP	a	0.092–0.23
**Quaternary ammonium compounds**						
Benzalkonium chloride	AP	AP	UR	AP	0.008–24.00	0.015–26.00
Benzalkonium saccharinate	AP	AP	UR	AP	a	0.10–0.20
Benzethonium chloride	NE	AP	NE	AP	a	0.28
Didecyldimethylammonium chloride	AP	AP	UR	AP	0.20–7.20	0.003–21.05
Didecylmethylpoly(oxethyl) ammonium propionate	AP	NE	UR	NE	a	b
Didecylmethylammonium carbonate/bicarbonate	NE	AP	NE	AP	a	0.0369–1.38
**Silver and derivates**						
Silver	AP	AP	UR	AP	0.004	0.003–0.01
Silver nitrate	AP	AP	UR	AP	a	0.016

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
