# Peer review of "A Critical Review of Disinfection Processes to Control SARS-CoV-2 Transmission in the Food Industry"

_foods, 2021, doi:10.3390/foods10020283_

Round 1

Reviewer 1 Report

This is an interesting review in which the authors describe the disinfection processes to control SARS-CoV-2 transmission in the food industry. The review is well written. However, I could not understand the necessity of Fig. 2.

Author Response

Dear rewiever,

Thank you a lot for your good rating and kind words about our paper. On second thoughts, we are agreeing with your suggestion about the lack of necessity of the Figure 2 so we have decided to remove from this new version.

Reviewer 2 Report

The manuscript presents an overview of the existing literature on the disinfection process for the control of SARS-CoV-2 transmission in the food industry and analyzes variables such as the type and concentration of the active substance, in order to ensure the inactivation of SARS. -CoV-2 and other usual pathogens and deteriorating microorganisms minimizing possible side effects on the environment and on animal and human health.

The review is well written and well organized. The information presented are new and interesting and the conclusions are supported by the data. Finally, the quality of the English is adequate.

I have only two minor revisions to suggest, as detailed below:

Line 210:  please remove the sentence highlighted in yellow (probably a note from one of the authors): “Almost ready after one rewriting more from here”

In my opinion, the authors should detail the caption of Figure 2 (Susceptibility level against disinfection of some foodborne pathogens. Adapted from [17]) a little better because the figure is not immediately understandable.

Author Response

Dear rewiever,

Thank you a lot for your good rating and kind words about our paper.  About your corrections:

“Line 210:  please remove the sentence highlighted in yellow (probably a note from one of the authors): “Almost ready after one rewriting more from here””

Yes, it is a note from one of the authors that we unfortunately forgot to remove before the first submission. We have fixed this issue in the new version of manuscript.

“In my opinion, the authors should detail the caption of Figure 2 (Susceptibility level against disinfection of some foodborne pathogens. Adapted from [17]) a little better because the figure is not immediately understandable.”

After your opinion and the words from other of the rewievers questioning the necessity of this figure, we have reconsidered to remove the Figure 2 in the revised version of the manuscript.

Reviewer 3 Report

A well-written report. I see some useful information presented especially the tables are designed very nicely and best form of representation of data. 

Author Response

Dear rewiever,

Thank you a lot for your good rating and kind words about our paper.

Reviewer 4 Report

Dear Authors,

The paper is interesting and usefull, especially from a practical point of view for the food sector.  In the conclusions I would  give more emphasis to the impact of substances in a one health perspective on the environment (man / food)

Some minor correction below.

line 6  substanceS

lines 73 74 However, the lack of a standardized methodology difficulties comparing the stability of the different viruses on same kind of surface" please rewrite the sentence

line 90 D isinfectants 

Author Response

Dear rewiever,

Thank you a lot for your good rating and kind words about our paper.  About your corrections:

“…line 6  substanceS”

“..line 90 D isinfectants” 

Both mistake have been fixed in this new version.

“…lines 73 74 However, the lack of a standardized methodology difficulties comparing the stability of the different viruses on same kind of surface" please rewrite the sentence.”

We have rewrite this sentence as follows for a better understanding.

"However, the lack of a standardized methodology makes complicated the comparison of literature results"